# Crystalline Hydrate Dehydration Sensing Based on Integrated Terahertz Whispering Gallery Mode Resonators

**DOI:** 10.3390/s22239116

**Published:** 2022-11-24

**Authors:** Zhibo Hou, Shixing Yuan, Wentao Deng, Jiahua Cai, Yaqin Qiu, Yunong Zhao, Ziwei Wang, Liao Chen, Huan Liu, Xiaojun Wu, Xinliang Zhang

**Affiliations:** 1Wuhan National Laboratory for Optoelectronics and School of Optical and Electronic Information, Huazhong University of Science and Technology, Wuhan 430074, China; 2School of Electronic and Information Engineering, Beihang University, Beijing 100191, China; 3Optics Valley Laboratory, Wuhan 430074, China

**Keywords:** terahertz (THz), whispering gallery mode (WGM), crystalline hydrate, dehydration, sensing

## Abstract

Water molecules play a very important role in the hydration and dehydration process of hydrates, which may lead to distinct physical and chemical properties, affecting their availability in practical applications. However, miniaturized, integrated sensors capable of the rapid, sensitive sensing of water molecules in the hydrate are still lacking, limiting their proliferation. Here, we realize the high-sensitivity sensing of water molecules in copper sulfate pentahydrate (CuSO_4_·5H_2_O), based on an on-chip terahertz whispering gallery mode resonator (THz-WGMR) fabricated on silicon material via CMOS-compatible technologies. An integrated THz-WGMR with a high-*Q* factor of 3305 and a resonance frequency of 410.497 GHz was proposed and fabricated. Then, the sensor was employed to distinguish the CuSO_4_·xH_2_O (x = 5, 3, 1). The static characterization from the CuSO_4_·5H_2_O to the copper sulfate trihydrate (CuSO_4_·3H_2_O) experienced blueshifts of 0.55 GHz/μmol, whereas the dehydration process of CuSO_4_·3H_2_O to copper sulfate monohydrate (CuSO_4_·H_2_O) exhibited blueshifts of 0.21 GHz/μmol. Finally, the dynamic dehydration processes of CuSO_4_·5H_2_O to CuSO_4_·3H_2_O at different temperatures were monitored. We believe that our proposed THz-WGMR sensors with highly sensitive substance identification capabilities can provide a versatile and integrated platform for studying the transformation between substances, contributing to hydrated/crystal water-assisted biochemical applications.

## 1. Introduction

Water molecules exist in many substances, such as copper sulfate pentahydrate (CuSO_4_·5H_2_O) [1,2], α-lactose monohydrate (C_12_H_22_O_11_·H_2_O) [3], and sodium thiosulphate pentahydrate (Na_2_S_2_O_3_·5H_2_O) [4,5,6]. Hydrate formation or dehydration of specific hydrates may produce a great impact on material properties, such as elastic properties [7,8,9], thermal structures [10,11,12], rheological properties [10,13,14,15], electrical properties [16], and activation energies [17]. The changes in the hydration state of crystalline compounds are usually unavoidable throughout the manufacturing process; hence, highly sensitive sensors for hydrates and hydration/dehydration monitoring are urgently required. At present, single-crystal X-ray diffraction (SCXRD) [18], powder XRD [19], infrared spectroscopy (IR) [20,21], Raman spectroscopy [22,23], nuclear magnetic resonance (NMR) spectroscopy [24], thermogravimetric analysis (TGA) [25,26,27,28], and corresponding auxiliary technology are commonly used to determine the content of crystal water. Compared with power XRD, SCXRD has the advantage of unambiguous peak indexing; however, it requires a single crystal of an adequate quality and size. IR and Raman spectroscopy fulfill the investigation of the vibrational mode, but the resolution is the limitation. NMR spectroscopy has the disadvantage of substantial cost. TGA requires a relatively large amount of samples (~mg) and a long time (~10 min) to perform. Therefore, a sensor with high accuracy and miniaturization is needed for crystal water sensing.

Terahertz (THz) sensing has received more and more attention in recent years. Most studies use terahertz time-domain spectroscopy (TDS) [29,30,31,32]. These kinds of methods have disadvantages such as demanding the preparation of sample tablets and relatively low-detection sensitivity. Terahertz resonators can be used for sensing and have the advantages of high accuracy and integration. Similar methods mainly focus on the terahertz whispering gallery mode resonators (THz-WGMRs) [33,34,35,36,37], metasurfaces [38,39,40,41], and photonic crystals [42,43,44]. THz-WGMRs possess high sensitivity because of high Q factors. The higher the Q implies the narrower the resonance dip in the transmission spectrum. High-Q microresonators can confine radiation with exquisitely low losses. Such low losses give rise to ultra-narrow resonance features that imply very fine frequency selectivity. Moreover, even a slight change in the surrounding medium will perturb the resonances to a measurable extent, allowing for environmental sensitivity [45] Therefore, THz-WGMRs have been employed in relevant research works in the fields of water vapor concentration measurement [45], particle distance sensing [46], and so on. However, THz-WGMRs have not been reported on crystal water sensing.

In this work, we designed an experiment to study the influence of CuSO_4_·xH_2_O (x = 5, 3, 1) on the THz spectrum of the THz-WGMR. Water molecules in CuSO_4_·5H_2_O combine in different chemical combinations. Investigating their dehydration process provides an auxiliary means to comprehend the chemical structure [17]. In addition, CuSO_4_·5H_2_O is rich in crystal water, and the temperature required for material transformation is easy to approach (less than 100 °C) [21,30]. Furthermore, the melting point is higher than the dehydration temperature, and consequently, the distribution of the substances maintains stability. The effect of crystal water on the refractive index and absorption coefficient of the material was analyzed. The proposed THz-WGMRs could distinguish CuSO_4_·xH_2_O (x = 5, 3, 1) with enhanced sensitivity and efficiency. The detection sensitivities of the THz-WGMR for CuSO_4_·5H_2_O, copper sulfate trihydrate (CuSO_4_·3H_2_O), and copper sulfate monohydrate (CuSO_4_·H_2_O) were 1.32, 0.77, and 0.56 GHz/μmol, respectively. Finally, we used this sensor to monitor the dehydration process at different temperatures.

## 2. Design of Devices and Experiments

The THz-WGMR, shown in Figure 1a, consists of a straight coupling waveguide and ring resonator, and the WGMs are excited through the terahertz waves in the straight waveguide. First, the THz wave is coupled into the straight waveguide. Then, the THz wave is coupled into the ring resonator where it is transmitted to the coupling region between the ring resonator and straight waveguide. Finally, the THz waves pass through the sample and export at the other end of the straight waveguide. The central resonant frequency *f* of the ring resonator can be expressed as
(1)$$f=mc/L_{eff}$$
in which *L_eff_* represents the roundtrip length of the microring, *m* is an integer, and *c* represents the speed of light in vacuum. The intensity transmissivity *T_t_* and the phase *ϕ* transmissivity of the microring resonator can be expressed as [47]
(2)$$T_{t}=\frac{r^{2}+a^{2}-2ra\cos\varphi }{1+r^{2}a^{2}-2ra\cos\varphi }$$
(3)$$\phi =\varphi +\pi +\tan^{-1}\frac{r\sin\varphi }{a-r\cos\varphi }+\tan^{-1}\frac{ar\sin\varphi }{1-ar\cos\varphi },$$
respectively, where *r* represents the self-coupling coefficient, *a* is the single-pass amplitude transmission coefficient of the ring resonator, and *φ* represents the single-pass phase shift of the ring. When the self-coupling coefficient *r* is equal to the single-pass amplitude transmission coefficient *a*, the THz-WGMR works in the critical coupling state, indicating the sensor exhibits the highest sensitivity to the surrounding environment.

The measurements were performed using a vector network analyzer (VNA) (Ceyear 3649B) with a frequency range of 0.325~0.5 THz, as shown in Figure 1b. The inner hole size of the metal waveguide was 508 × 254 μm^2^ (WR-2.2). A linearly polarized THz radiation was generated from the emitter and received by the receiver. The VNA obtained the intensity transmission spectrum and phase spectrum of the THz-WGMR in the target frequency band using frequency scanning.

In this work, a high-*Q* THz-WGMR based on high-resistance float-zone silicon (HRFZ-Si) using traditional CMOS-compatible technologies was proposed and fabricated [46], as shown in the inset of Figure 1a. The radius *r* of the ring resonator was 5.05 mm, and the width *w* of the straight waveguide was 304.02 μm. The width *w* of the ring waveguide was 298.83 μm. The total thickness of the chip was 120 μm, and the thickness of ridge *H_1_* was 63.13 μm. The gap *g* between the straight waveguide and the microring was 36.33 μm. The proposed THz-WGMR suppresses high-order modes, guaranteeing the purity of the spectrum.

We used the THz-WGMR to measure the change in the complex refractive index of the substance as the sensing theory, for which different substances usually have different complex refractive indices which make different effects on the resonance dip of the resonator. In detail, the substance overlapping with the evanescent field will change the mode extinction coefficient and mode refractive index corresponding to the change in the single-pass amplitude transmission coefficient *a* and the roundtrip length *L_eff_*, which leads to the change in the transmissivity *T_t_* and resonance frequency *f* of the THz-WGMR, respectively.

In this experiment, the type (CuSO_4_·5H_2_O, CuSO_4_·3H_2_O, and CuSO_4_·H_2_O) of hydrate and the amount of CuSO_4_·xH_2_O (x = 5, 3, 1) covering the THz-WGMR were independent variables. The transmission spectrum of the THz-WGMR was measured to demonstrate the relationship between the complex refractive indices and the crystal water content. In addition, the dynamic dehydration and transformation from CuSO_4_·5H_2_O to CuSO_4_·3H_2_O at different temperatures was observed. To quantify the sample, the supersaturation CuSO_4_ solution with a concentration of 2 mol/L was first prepared. Then, the solution was transferred using a pipette (0.1~2.5 μL) quantitatively every time and then cooled to crystallize to obtain CuSO_4_·5H_2_O.

## 3. Results and Discussion

### 3.1. Water Content Identification of CuSO_4_·xH_2_O (x = 5, 3, 1)

To verify the feasibility of the THz-WGMR for crystal water sensing, the device was utilized to distinguish CuSO_4_·xH_2_O (x = 5, 3, 1) using complex refractive indices measurement. We observed the extinction ratio (ER) and the frequency shift of the resonance dip, which corresponded to the absorption coefficient and refractive index of the CuSO_4_·xH_2_O (x = 5, 3, 1), respectively. The transmission spectra of the THz-WGMR with CuSO_4_·xH_2_O (x = 5, 3, 1) on the ring resonator were measured. 

To determine the change in the hydration state of the crystalline compounds, the experimental procedure was specially designed. The specific operations are shown in Appendix A. Briefly, we first prepared 2 mol/L of supersaturated CuSO_4_ solution at 80 °C, then quantitatively transferred the liquid to the surface of the ring resonator with a pipette, and finally, crystallized the solution at 25 °C to precipitate the CuSO_4_·5H_2_O crystal based on a temperature control plate under the chip. Here, one drop of 0.5 μL of the supersaturated CuSO_4_ solution was placed above the ring resonator, as shown in Figure 1a. After cooling and crystallizing, one dot of 1 μmol of CuSO_4_·5H_2_O was successfully obtained above the ring resonator. The above transfer method keeps the amount and distribution of CuSO_4_·5H_2_O stable as far as possible in repeat experiments. Then, the method of heating dehydration was used to realize the transformation from CuSO_4_·5H_2_O to CuSO_4_·3H_2_O and finally, to CuSO_4_·H_2_O, as detailed in Appendix A. In general, CuSO_4_·5H_2_O will gradually transform into CuSO_4_·3H_2_O when the heating temperature is higher than 50 °C, and it will gradually transform into CuSO_4_·H_2_O when the heating temperature is higher than 90 °C [21,30]. After measuring the spectrum of the THz-WGMR with the CuSO_4_·5H_2_O, the heating temperature was adjusted to 60 °C to fulfill the transformation from CuSO_4_·5H_2_O to CuSO_4_·3H_2_O, then the spectrum of the THz-WGMR with the CuSO_4_·3H_2_O was recorded. Similarly, the heating temperature was adjusted to 100 °C to fulfill the transformation from CuSO_4_·3H_2_O to CuSO_4_·H_2_O, and the spectrum of the THz-WGMR with the CuSO_4_·H_2_O was finally acquired. This method possesses a high accuracy in the experimental results by avoiding the complex operations of the addition and removal of substances during the measurement. The ambient temperatures will change the ER and resonant frequency of the resonator which affects the experimental results. Therefore, to keep the results reasonable, all the transmission spectra were recorded at a temperature of 25 °C. The time for the hydration of the CuSO_4_·H_2_O and CuSO_4_·3H_2_O by moisture adsorption at 25 °C beyond one hour was far larger than the operational time, which was approximately 5 min to adjust the temperature and the time to acquire and store the data (10 s), as illustrated in Appendix A. The transformation from CuSO_4_·H_2_O to CuSO_4_·3H_2_O and from CuSO_4_·3H_2_O to CuSO_4_·5H_2_O during the operational process was less than one percent.

The intensity transmission spectra of the THz-WGMR covered by one dot of 1 μmol of CuSO_4_·xH_2_O (x = 5, 3, 1) are illustrated in Figure 2a. The insets illustrate the measured intensity spectrum and the phase profile together with the fit lines, which demonstrate that the THz-WGMR works in the critical coupling state with the *Q* value of 3305 at the resonant frequency of 410.497 GHz. From the results, the substances increase the transmission loss of the THz-WGMR which pushes the coupling state away from the critical coupling state, therefore, leading to the decrease in the ER. In addition, the more the crystal water content implies the smaller the ER of the resonance dip. Roughly, the water content possesses a negative correlation with the ER, implying a positive correlation with the absorption coefficient of the substances. Focusing on the frequency shift, as the effective roundtrip length of the THz-WGMR increases with the effect of the substances, the resonant frequency of the THz-WGMR decreases. The more the crystal water content implies the greater the frequency shifts of the resonance dip. Roughly, the water content possesses a positive correlation with the frequency shift, implying a positive correlation with the refractive index.

Further, from the transmission spectra in Figure 2a, the ER will change greatly when the transmission loss of the THz-WGMR changes, especially for the THz-WGMR which is close to the critical coupling state. Hence, these sensors possess advantages in tiny substance detection. The ERs and frequency shifts were extracted from Figure 2a, and the experiments were repeated three times. Figure 2b illustrates the mean values and standard deviations of the ERs and frequency shifts caused by one dot of 1 μmol of CuSO_4_·xH_2_O (x = 5, 3, 1). As shown in Figure 2b, the higher water contents imply lower ERs and higher frequency shifts. Focusing on the standard deviation of the ER, the result from the CuSO_4_·H_2_O possesses the largest value, which is 2.8 dB, per the above argument; as shown in Figure 2a, the blue curve (CuSO_4_·H_2_O) is closer to the critical coupling state and, therefore, possesses the most intense change in the ER from the deviation of the amount and distribution of the substances. However, focusing on the standard deviations of the frequency shifts, the result from the CuSO_4_·5H_2_O possesses the largest value, which is 0.16 GHz, resulting from the largest refractive index, for which the deviation of the amount and distribution of the CuSO_4_·5H_2_O on the ring resonator will bring remarkable deviations. Nevertheless, these sensors distinguish CuSO_4_·xH_2_O (x = 5, 3, 1) clearly both in the ERs and frequency shifts corresponding with the substance absorption coefficients and refractive indexes. It is noteworthy that these sensors exhibit application potential in trace amount crystalline hydrate sensing.

### 3.2. Sensitivity Characterization

To demonstrate the sensitivity of the THz-WGMR for CuSO_4_·xH_2_O (x = 5, 3, 1), different amounts of CuSO_4_·xH_2_O (x = 5, 3, 1) were used to perform the sensing and characterization of the sensitivity.

In this experiment, six drops of supersaturated CuSO_4_ solution with a concentration of 2 mol/L and a volume of 0.1 μL per drop were first transferred onto the surface of the ring resonator discretely using a pipette and then cooled to crystallize at 25 °C to obtain six dots of 0.2 μmol per dot of CuSO_4_·5H_2_O, as shown in Appendix A. After that, one dot of 0.2 μmol of CuSO_4_·5H_2_O was removed with a cotton swab step by step; meanwhile, the spectrum of the THz-WGMR was recorded. Next, the transmission spectra of the THz-WGMR covered with 1.2, 1, 0.8, 0.6, 0.4, 0.2, and 0 μmol of CuSO_4_·5H_2_O were measured, respectively, and the ERs and frequency shifts of the resonance dip were extracted, as shown in Figure 3. According to the above recommendation in Section 3.1, the heated dehydration process should fulfill the transformation from CuSO_4_·5H_2_O to CuSO_4_·3H_2_O and from CuSO_4_·3H_2_O to CuSO_4_·H_2_O, then follow the same removal operation to obtain the ERs and frequency shifts caused by the different amounts of CuSO_4_·3H_2_O and CuSO_4_·H_2_O. We speculated that the variety of the volume of the drops and the overlapping areas between the substances and the evanescent fields of the waves in the THz-WGMR were the main sources of the deviations. We estimated the deviation of the overlapping areas, and the statistical result was ±6.5%, as demonstrated in Appendix A.

The experimental results (dots) were fitted using the formula (2). As shown in Figure 3, the ERs decrease with the increase in the amount of CuSO_4_·xH_2_O (x = 5, 3, 1) due to the increase in the extinction coefficients which push the THz-WGMR away from the critical coupling state. However, the frequency shifts increase linearly owing to the linear increase in the effective roundtrip length. We observed that the fitting values and the ERs mismatch at the amount of 0 μmol, mainly due to the residues of CuSO_4_·xH_2_O (x = 5, 3, 1) affecting the THz-WGMR when the amount is 0. Furthermore, the ER is sensitive to the surrounding environment when the THz-WGMR works near the critical coupling state, leading to an obvious deviation. The residues of CuSO_4_·xH_2_O (x = 5, 3, 1) arise from the operation procedures. As demonstrated in Appendix A, six dots of 0.2 μmol per dot of CuSO_4_·xH_2_O were transferred onto the surface of the ring resonator discretely, and then they were removed one by one with cotton swabs. Therefore, there inevitably were residues on the surface.

According to the results in Figure 3, the relationship between the amount of CuSO_4_·xH_2_O (x = 5, 3, 1) and frequency shifts maintain high linearity in a relatively large range, which makes it an appropriate index to define the sensing sensitivity of the sensor. The ERs demonstrate a nonlinear relationship with the amount of CuSO_4_·xH_2_O (x = 5, 3, 1) in the experimental range, resulting from the nonlinear relationship between the transmissivity *T_t_* and the single-pass amplitude transmission coefficient *a*. However, they can be approximated as a linear relationship in a small range. By calculating the slopes of the blue fitting lines, the sensitivities of the sensor for CuSO_4_·xH_2_O (x = 5, 3, 1) were obtained as 1.32, 0.77, and 0.56 GHz/μmol, respectively.

In fact, in this experiment, to ensure a high-crystallization rate, a high concentration of 2 mol/L of supersaturated CuSO_4_ solution was used, which led to the minimum transferable amount of 0.2 μmol of the substance in this experiment. However, it was found that 0.2 μmol of substances does not completely interact with the evanescent field of the THz wave around the THz-WGMR, implying that the practical detection sensitivity of the sensor is higher than the results in the experiment.

To verify the experimental results, the effects of different amounts of CuSO_4_·xH_2_O (x = 5, 3, 1) on the transmission spectra of the THz-WGMR were simulated and analyzed. To establish the simulated model, the complex refractive indices of CuSO_4_·xH_2_O (x = 5, 3, 1) were required to measure first. The processes were as follows. Firstly, the CuSO_4_·xH_2_O (x = 5, 3, 1) tablets were prepared using a tablet machine, then their transmission spectra were measured using the THz-TDS. The thicknesses and transmission spectra of the tablets were acquired and used to calculate the refractive indexes and extinction coefficients. The details are shown in Appendix A. The results are shown in Figure 4a, and we noticed that the crystal water content of the CuSO_4_ crystal possesses a positive correlation with its refractive index and extinction coefficient, which is consistent with the experimental results. In detail, the refractive indexes of CuSO_4_·xH_2_O (x = 5, 3, 1) at 410.5 GHz are 2.4, 2.3, and 2.1, respectively, and the extinction coefficients are 0.053, 0.041, and 0.037, respectively.

After measuring the size of the devices and samples, a two-dimensional axisymmetric model was established in COMSOL, as detailed in Appendix A, and then the mode refractive indexes and mode extinction coefficients of the THz wave interacting with the CuSO_4_·xH_2_O (x = 5, 3, 1) were simulated. According to the data from Figure 3, the mode refractive indexes and mode extinction coefficients of the THz waves in the experiments can be calculated too. Comparing the calculated results with the simulated one, the relative errors of the mode refractive indexes caused by CuSO_4_·xH_2_O (x = 5, 3, 1) are 0.021, 0.003, and 0.011, respectively and the relative errors of the mode extinction coefficients caused by CuSO_4_·xH_2_O (x = 5, 3, 1) are 0.19, 0.31, and 0.28, respectively. The detailed values are listed in Appendix A. Interestingly, the mode refractive indexes from the two methods correctly match each other. However, the mode extinction coefficients possess large relative errors, which result from the reflective loss from the samples which is not considered in the simulations, and unfortunately, the surface morphology and porosity of the substances are hard to obtain; hence, the detected substances are not perfectly reproduced in the simulation. Nevertheless, we simulated the effects of the different amounts of CuSO_4_·xH_2_O (x = 5, 3, 1) on the ERs and frequency shifts of the resonance dip, and the results are shown in Figure 4b. It is noteworthy that the ERs demonstrate a nonlinear relationship with the amount of the CuSO_4_·xH_2_O (x = 5, 3, 1), and the frequency shifts increase linearly with the amount of the CuSO_4_·xH_2_O (x = 5, 3, 1), which is qualitatively consistent with the experimental results in Figure 3.

### 3.3. Monitoring of Dynamic Dehydration from CuSO_4_·5H_2_O to CuSO_4_·3H_2_O at Different Temperatures

Finally, the processes of CuSO_4_·5H_2_O dehydration and transformation into CuSO_4_·3H_2_O at different temperatures are exhibited in Figure 5. First, the heating temperatures were set as 60, 65, and 70 °C, respectively, and then the CuSO_4_·5H_2_O particles were directly transferred above the THz-WGMR. As demonstrated in Figure 5, the ER and frequency shift of the resonance dip increase over time until a steady state is reached. This phenomenon arises as the extinction coefficient and refractive index decrease when the CuSO_4_·5H_2_O dehydrates into the CuSO_4_·3H_2_O. The decrease in the extinction coefficient makes the THz-WGMR get closer to the critical coupling state which explains the increase in the ER. In addition, the decrease in the refractive index accumulates the frequency shift. Finally, the dynamic process reaches a steady state when the CuSO_4_·5H_2_O completely transforms into CuSO_4_·3H_2_O.

The results in Figure 5 demonstrate that the higher the temperature is, the less time there is to reach the steady state, indicating a higher dehydration rate with the augments in temperature. We speculate that the dehydration rate mainly depends on the temperature in this experiment. The ERs and frequency shifts increase synchronously and then reach steady states almost at the same time. During the experiment, we noticed that there was a clearer temperature fluctuation with a higher heating temperature, and the temperature fluctuation mainly affected the frequency shifts. Hence, taking the time when ER reaches the steady state as the observation index, the transformation time can be expressed more accurately, as marked in Figure 5.

Generally, this experiment demonstrates that these sensors possess the ability to dynamically monitor the dehydration process of substances, providing a method for analysis and the investigation of the dynamic change in substances.

## 4. Conclusions

In conclusion, an on-chip THz-WGMR was proposed and fabricated, enabling sensitive and rapid sensing of crystal water. The device was applied to measure different amounts of CuSO_4_·xH_2_O (x = 5, 3, 1). The detection sensitivities of the THz-WGMR for CuSO_4_·xH_2_O (x = 5, 3, 1) were 1.32, 0.77, and 0.56 GHz/μmol, respectively. The experimental results are in good agreement with the simulation results qualitatively, indicating that the extinction coefficient and refractive index increase with the increase in crystal water content. This difference in the determination parameters can be used to identify specific substances with different crystal water content. The experiment also demonstrates the dehydrated process from CuSO_4_·5H_2_O to CuSO_4_·3H_2_O with increased time at different temperatures, suggestive of the dynamic monitoring capability of the proposed device. The results show that a higher temperature implies a higher dehydration rate. It is noteworthy that the ability of tiny substance detection from the THz-WGMR was verified by the experiment, and its feasibility in the dynamic sensing of crystal water was proved. We believe that these sensors will provide a powerful tool for material analysis and the dynamic monitoring of biochemical reactions.

## Figures and Tables

**Figure 1 sensors-22-09116-f001:**
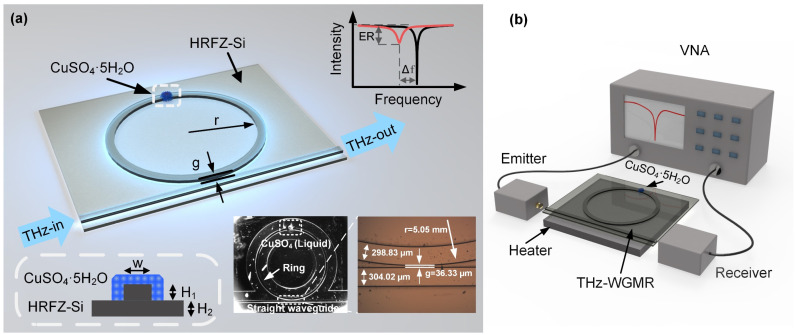
The schematic diagram of the terahertz whispering gallery mode resonator (THz-WGMR) together with its sensing principle and the diagram of the experimental setup. (**a**) The schematic diagram of the THz-WGMR. The inset in the lower left shows the cross-section of the ridge waveguide. The spectra in the upper right illustrate the principle of sense. The photographs in the lower right show the real chip which consists of a straight coupling waveguide and a ring resonator. The supersaturated CuSO_4_ solution of 0.5 μL covers the ring resonator. The optical microscopic image on the right shows the coupling region of the THz-WGMR. (**b**) The illustration of the experimental setup. The heating plate under the chip is used to adjust and maintain the temperature. The end faces of the straight waveguide are aligned with the emitter and receiver, and the transmission spectrum is shown on the screen of the vector network analyzer (VNA).

**Figure 2 sensors-22-09116-f002:**
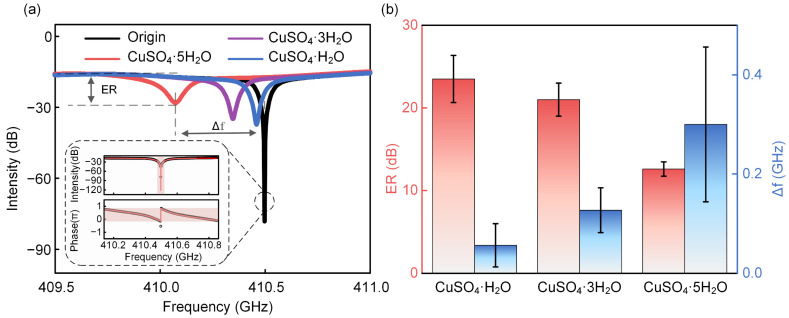
The spectra of the THz-WGMR and the experimental results for CuSO_4_·xH_2_O (x = 5, 3, 1). (**a**) The intensity spectra of the THz-WGMR covered with one dot of 1 μmol CuSO_4_·xH_2_O (x = 5, 3, 1). The insets are the intensity spectrum and phase profile of the THz-WGMR. The black circles and red lines in insets represent the experimental and analytical results, respectively. The intensity spectrum of the THz-WGMR illustrates the resonance frequency of 410.497 GHz and the quality (Q) value of 3305. The phase profile demonstrates a π shift, indicating the THz-WGMR works in the critical coupling state. (**b**) The extinction ratios (ERs) and frequency shifts of the resonance dip caused by one dot of 1 μmol (0.5 µL) CuSO_4_·xH_2_O (x = 5, 3, 1).

**Figure 3 sensors-22-09116-f003:**
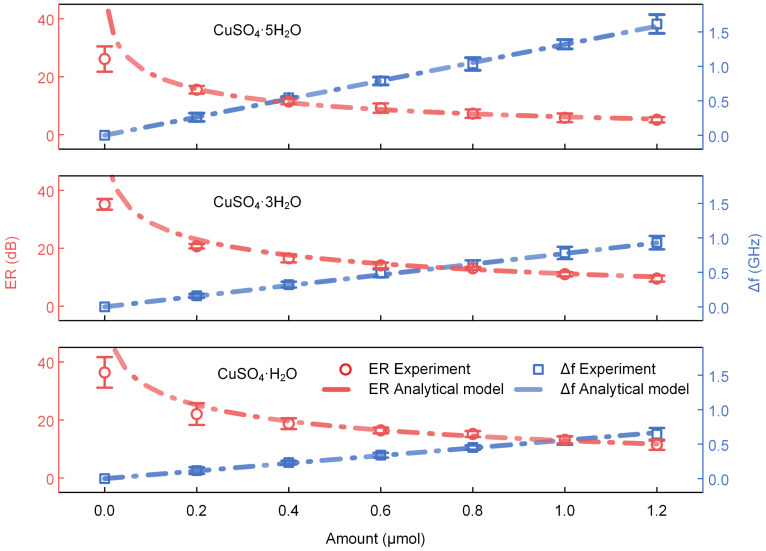
The experimental results (dots) and fitting results (lines) for different amounts of CuSO_4_·xH_2_O (x = 5, 3, 1). The red and blue dots (curves) represent the ERs and frequency shifts of the resonance dip, respectively.

**Figure 4 sensors-22-09116-f004:**
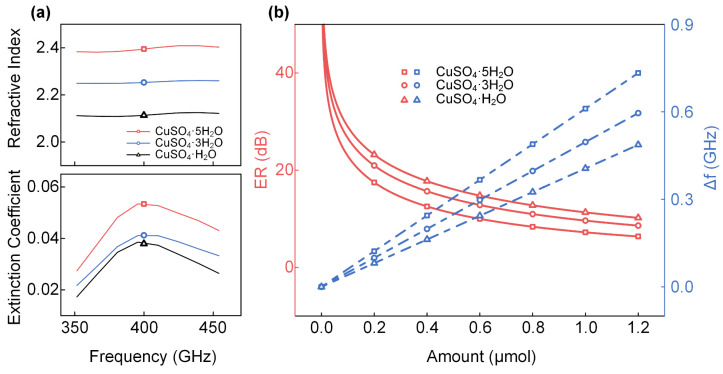
The complex refractive indices of CuSO_4_·xH_2_O (x = 5, 3, 1) and the simulated sensing results for different amounts of CuSO_4_·xH_2_O (x = 5, 3, 1). (**a**) Refractive indexes and extinction coefficients of CuSO_4_·xH_2_O (x = 5, 3, 1) with the frequency from 350 to 450 GHz. (**b**) The ERs (red) and frequency shifts (blue) of the resonance dip.

**Figure 5 sensors-22-09116-f005:**
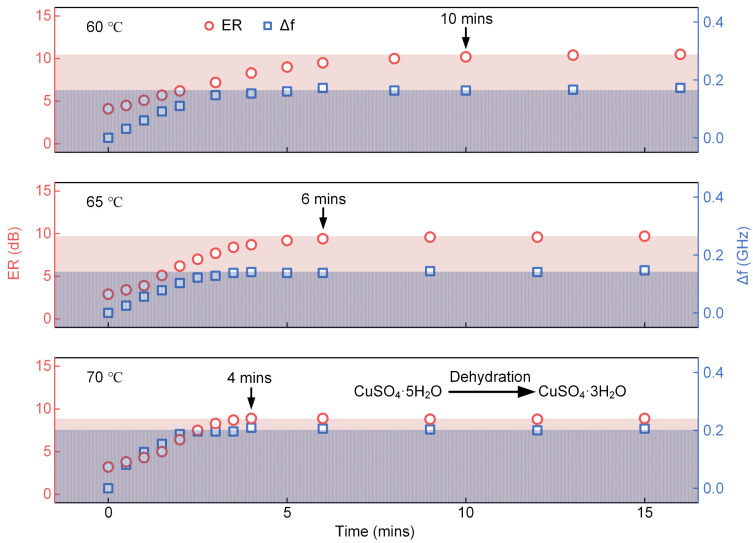
The dynamic dehydration from CuSO_4_·5H_2_O to CuSO_4_·3H_2_O at 60, 65, and 70 °C, respectively. The ERs (red circles) and frequency shifts (blue squares) of the resonance dip along the time are demonstrated.

## Data Availability

Data is available upon request. Please contact the corresponding author.

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
