# Peer review of "Crystalline Hydrate Dehydration Sensing Based on Integrated Terahertz Whispering Gallery Mode Resonators"

_sensors, 2022, doi:10.3390/s22239116_

Round 1

Reviewer 1 Report

This manuscript presents an integrated terahertz WGMR-based sensing of dehydration of crystalline hydrates. Since WGMR has a high Q value to enhance the interaction of CuSO4 5H2O with terahertz waves, the sensitivity performance is very good. The authors have done an excellent manuscript preparation and the experimental results are detailed and agree well with the simulation results. Given the range of excellent experimental results, the manuscript is accepted for publication with minor revisions. Authors should make the following comments:

- High Q WGMR can improve the sensitivity of the sensor. The authors should discuss the relationship between Q and sensitivity.

- In Figure 3, when the equivalent is 0, the experimental data deviates from the theoretical value, please explain.

- In Figure 5, the 70° time should be 4 minutes better than the 3.5 minutes, the 3.5 minute data seems to be in a slow upward trend.

Reviewer 2 Report

Crystalline hydrate dehydration sensing based on integrated terahertz whispering gallery mode resonators’ by Hou et. al. describes the technological application of terahertz whispering gallery mode resonator to detect concentration of molecular water in CuSO4 compounds. Results are well described and experiments were compared to simulated results to establish reliability of the work. So, I recommend the article to be published with minor changes (see below) in Sensors, since it would be a great benefit for the researchers to the scope of the Sensors.

Please consider following comments

1.       Introduction: It would be good to mention why authors picked CuSO4 and two other hydrates from ‘many substances’.

2.       Results 3.1 line 159-161: Please add proper references or supporting materials to validate your statement.

Reviewer 3 Report

The authors present an integrated sensor based on a THz WGM microring resonator on silicon capable of dynamically monitoring the dehydration process of substances. The drop of the sample solution is transferred by a pipette and placed over a portion of the ring wall.

The manuscript is well structured and written, the reader can easily follow the work done by authors but the lack of some detailed descriptions does not allow to fully understand the procedures.

Furthermore, some transductive contributions that can strongly influence the accuracy and errors evaluation have not been quantified.

The manuscript is not ready to be accepted and it may be only after addressing the comments below:

1)      Row 139, “The specific operations are shown in Supplementary Note 1”; Row 148, “as detailed in Supplementary Note 2.”; Row 251 “The details are shown in Supplementary Note 3.” etc…: Please, could the authors provide these detailed descriptions?

2)      Row 217: The authors should quantify the variation of the drops volume and the relative variation of the ring portion touched by the drop, in order to distinguish their contributions (ER and shift) and to evaluate whether the introduced error does not compromise the value of the measurement.

Round 2

Reviewer 3 Report

The authors have correctly addressed the comments and improved the previous version of the manuscript.